# Sexual Dysfunction in Female Rectal and Anal Cancer Survivors: Pathophysiology, Clinical Management, and Integration into Survivorship Care

**DOI:** 10.3390/cancers17193150

**Published:** 2025-09-28

**Authors:** Denise Drittone, Monia Specchia, Eva Mazzotti, Federica Mazzuca

**Affiliations:** 1Medical Oncology Unit, Sant’Andrea Hospital in Rome, Via di Grottarossa 1035–1039, 00189 Rome, Italy; monia.specchia@uniroma1.it (M.S.); eva.mazzotti@ospedalesantandrea.it (E.M.); 2Oncology Unit, Department of Clinical and Molecular Medicine, Sant’Andrea University Hospital, Sapienza University of Rome, Via di Grottarossa 1035–1039, 00189 Rome, Italy; federica.mazzuca@uniroma1.it

**Keywords:** female sexual dysfunction, rectal cancer, anal cancer, pelvic surgery, radiotherapy, survivorship, quality of life, FSFI, psychosexual care, immunotherapy

## Abstract

This review looks at how treatments for rectal and anal cancers affect women’s sexual health, a topic that is often ignored in cancer care. While survival rates are improving, many women experience lasting problems such as pain during sex, loss of desire, and changes in intimacy. These issues are caused not only by physical effects of treatment, like tissue damage and hormonal changes, but also by emotional factors, including low self-esteem, body image concerns, and relationship difficulties. Current ways of measuring sexual health do not always reflect the real experiences of female survivors, and support services remain limited. We highlight the need for better tools, more consistent care, and greater attention to sexual well-being as an essential part of life after cancer.

## 1. Introduction

Colorectal cancer represents about 10% of global cancer cases, ranking second among women, who experience approximately 25% lower incidence and mortality than men [1]. Meanwhile, the incidence of anal cancer, driven mainly by HPV-related squamous cell carcinoma, is steadily rising worldwide, with the sharpest increases seen in women over 50 years [2].

Advancements in the treatment of rectal and anal cancers have significantly improved survival, shifting clinical attention toward long-term quality of life in female patients. Among the critical dimensions of survivorship, sexual function remains a key aspect of biopsychosocial well-being and is often under-recognized in clinical oncology.

Despite its potential impact on quality of life, female sexuality post-treatment remains under-investigated, with a scarcity of gender-specific data and validated tools for assessment. Furthermore, the area of sexual dysfunction is not a priority for the clinician’s assessments in the pre- and post-treatment phase.

Most literature focuses on male patients or includes mixed populations, thereby limiting its applicability to women specifically. Therefore, this review aims to synthesize and critically analyze available evidence on sexual dysfunction in women treated for rectal and anal cancers with surgery and/or chemoradiotherapy. It will explore pathophysiological mechanisms, clinical manifestations, assessment tools, and prevention or treatment strategies, emphasizing existing gaps and future research directions.

Sexual dysfunction after treatment for rectal and anal cancers can manifest as decreased libido, vaginal dryness, dyspareunia, difficulties with arousal or orgasm, and sexual avoidance [3]. Contributing factors include anatomical and neurological damage from surgery, side effects of pelvic radiotherapy, hormone alterations induced by chemotherapy (Figure 1), and psychological factors such as anxiety, depression, and altered body image.

This review aims to integrate clinical and practical perspectives by critically examining the limitations of current assessment tools and proposing potential pathways for the incorporation of sexual health into the comprehensive care of female patients who have completed treatment for rectal and anal cancer. Furthermore, this review seeks to outline strategies to bridge the gap between evidence synthesis and practical implementation in real-world oncology settings, thereby supporting a multidisciplinary approach to enhance sexual health and quality of life among female survivors of pelvic malignancies.

## 2. Physiology of Female Sexuality and Oncologic Impact

The pathophysiology of female sexual dysfunction appears more complex than that of males, involving multidimensional hormonal, neurological, vascular, psychological, and interpersonal aspects [4]. Female sexual arousal is triggered by sensory stimulation and central nervous system activation. The limbic system and hypothalamus coordinate autonomic, affective, and cognitive processes that lead to vasocongestion, vaginal lubrication, clitoral engorgement, and heightened genital sensitivity. These responses are mediated through sympathetic, parasympathetic, and somatic pathways, supported by key neurotransmitters such as dopamine, serotonin, and oxytocin. Hormonal fluctuations and psychological factors, including mood and relational context, further modulate desire, arousal, and sexual satisfaction [5].

Oncologic treatments for colorectal or anal cancer can disrupt female sexual function at multiple levels; bilateral oophorectomy or chemotherapy/chemoradiation–induced ovarian failure precipitates rapid hormonal withdrawal—predominantly estrogen and androgen deficiency—leading to vaginal atrophy, impaired lubrication, diminished libido, and dyspareunia [6]. Pelvic surgery, particularly total mesorectal excision and abdominoperineal resection, can damage the pelvic autonomic pathways (superior/inferior hypogastric plexus and pelvic splanchnic nerves) that mediate genital arousal and lubrication, leading to postoperative sexual dysfunction [7]. Furthermore, pelvic radiation damages the vaginal epithelium, microvasculature, and connective tissue, triggering inflammation, reduced perfusion/hypoxia, hyalinization, and fibrosis that thin the mucosa and diminish lubrication. These changes commonly lead to vaginal shortening and stenosis, reduced elasticity, and pain with intercourse [8].

The impact of these dysfunctions is significant: a systematic review and meta-analysis encompassing 35 studies reported that over 60% of female cancer patients experience sexual dysfunction, with average FSFI scores below 20 across various tumor types, including colorectal and gynecological cancers [9]. FSFI is a validated multidimensional questionnaire widely used to assess female sexual function across six domains: desire, arousal, lubrication, orgasm, satisfaction, and pain. Scores range from 2 to 36, with a total score below 26.55 generally indicating risk of sexual dysfunction. This instrument is frequently used in oncologic research to quantify sexual health outcomes objectively [10].

In a prospective study of colorectal cancer survivors undergoing preoperative radiotherapy, FSFI total scores dropped significantly from 18.5 to 10.8 post-treatment (*p* < 0.001), reflecting widespread sexual function impairment [11]. Similarly, women treated with pelvic radiotherapy showed markedly lower FSFI scores compared to healthy controls (mean 8.5 vs. 13.5; *p* = 0.049) [12]. In women treated for anal cancer, significant reductions in sexual desire and increased rates of dyspareunia have been consistently reported, with FSFI scores demonstrating marked deterioration across most quality-of-life domains, while relational satisfaction appeared relatively preserved [13].

Empirical data from a prospective cohort of colorectal/anal cancer survivors (*n* = 97) revealed that only approximately 50% remained sexually active post-treatment, among whom over 70% scored below the FSFI threshold for sexual dysfunction (total FSFI < 26.55). Median FSFI total scores were 21.8 within one year of treatment and remained low (median, 22.6) beyond two years; the desire subscale medians were around 3.0, significantly below the clinical cutoffs [14]. Other studies report postoperative dysfunction prevalence ranging from 19% to 62%, with 30–40% of previously sexually active patients ceasing sexual activity altogether [15].

Pelvic radiation exacerbates these dysfunctions: registry data from Norway show that women receiving surgery plus (chemo)radiation have significantly higher rates of vaginal dryness (50% vs. 24%), dyspareunia (35% vs. 11%), and vaginal shortening (35% vs. 6%) compared to surgery alone, despite no differences in sexual interest [16].

Psychosocial factors such as altered body image, fecal incontinence, or stoma presence add to the burden, though physiological impairment appears to be the predominant driver of sexual dysfunction [14].

According to the NCCN Survivorship Guidelines (Version 2.2024), sexual dysfunction in female cancer survivors is inherently multifactorial, arising from physiological (e.g., menopause, mucosal injury), psychological, interpersonal, and treatment-induced causes. The guidelines recommend comprehensive, multidisciplinary evaluation and intervention, including vaginal estrogen or lubricants, pelvic floor physical therapy, psychosocial or couples counseling, and lifestyle modifications. However, regenerative therapies like vaginal dilators or off-label medications currently lack strong supporting evidence [17].

Additionally, the psychological burden of infertility and body image changes following oncologic surgeries significantly affects sexual well-being and overall quality of life [18].

## 3. Rectal and Anal Cancer Surgery and Sexual Dysfunction

Surgical interventions, including low anterior resection (LAR), abdominoperineal resection (APR), and total mesorectal excision (TME), represent fundamental components in the management of locally advanced rectal cancer [19]. Pelvic surgeries risk injuring the autonomic nerves (hypogastric plexuses, pelvic splanchnic nerves), frequently resulting in bladder, bowel, and sexual dysfunction, which remains highly prevalent among women after rectal cancer surgery despite advances in nerve-sparing techniques [20,21]

During deep anterior resection in the small pelvis, the anterior dissection of the rectum represents a critical danger zone where pelvic autonomic nerves are particularly vulnerable. In women, injury at this stage may compromise parasympathetic fibers involved in genital vasocongestion, vaginal lubrication, and clitoral sensitivity, leading to dyspareunia and reduced sexual satisfaction postoperatively. The risk of such nerve damage increases with the depth of dissection, reflecting the challenges of surgical hemostasis in this anatomically constrained region [4,22].

During colorectal surgery, nerve injury risk is highest during ligation of the inferior mesenteric artery, posterior and lateral rectal mobilization, and deep anterior dissection near the prostate, potentially leading to autonomic dysfunction. Careful technique in these four key danger zones is essential to balance oncologic outcomes with nerve preservation [22]. Although APR remains the standard for low rectal tumors, it is associated with higher recurrence, poorer survival, and greater impacts on physical function, body image, and sexual health than LAR, underscoring the need for preoperative counseling and shared decision-making [23]. Also, concerning TME, while it improves local control in rectal cancer, it often results in sexual and urinary dysfunction due to intraoperative nerve injury [24].

Advanced age, low tumor location, preoperative radiotherapy, and undergoing APR, ISR, or Hartmann procedures are independent risk factors for postoperative sexual dysfunction after rectal cancer surgery, likely due to higher risks of pelvic autonomic nerve injury [25].

These aspects highlight the need for structured preoperative counseling, nerve-sparing surgical strategies, and post-treatment sexual health support to optimize survivorship care in women undergoing rectal and anal cancer surgery.

## 4. Radiotherapy/Chemoradiotherapy: Late Effects on Sexual Function

Pelvic radiotherapy and chemoradiotherapy, key treatments for rectal and anal cancers, can lead to long-term female sexual dysfunction due to progressive tissue damage occurring months to years after treatment. Radiation-induced fibrosis, with excessive collagen deposition and scarring in the vaginal wall, pelvic floor muscles, and surrounding tissues, reduces tissue elasticity and compliance, contributing to pain and sexual difficulties [26,27].

Vaginal stenosis is a frequent late complication after pelvic radiotherapy, with incidence rates up to 88%. The underlying mechanisms include radiation-induced fibrosis, reduced vascularization, and epithelial atrophy. Preventive measures such as vaginal dilator use and maintaining sexual activity are recommended to reduce risk [8]. Radiation-induced peripheral neuropathy is a rare but increasing late effect in long-term cancer survivors, typically appearing years after treatment. It results from nerve compression due to fibrosis, direct nerve damage, and vascular ischemia. Incidence has decreased with modern radiotherapy, but when it occurs, it is often progressive and impacts quality of life [28]. This can be another critical mechanism of sexual dysfunction.

Modern radiotherapy techniques have evolved to minimize these adverse effects while maintaining oncologic efficacy. The use of intensity-modulated radiotherapy (IMRT), well-established in gynecologic cancers, is equally crucial in rectal and anal cancers to limit radiation exposure to the bladder, rectum, and pelvic floor. By reducing dose to these structures, IMRT may help preserve pelvic function and mitigate sexual dysfunction, improving survivors’ long-term quality of life [29].

Overall, recognizing and managing these late effects is essential to optimize survivorship care and sexual health in women undergoing pelvic RT or CRT.

Chemotherapeutic agents, particularly alkylating agents such as cisplatin and 5-fluorouracil, can damage reproductive tissues and alter hormonal levels, leading to premature menopause, reduced sexual desire, and pain during intercourse [30]. Additionally, paclitaxel, a commonly used taxane, has been shown to contribute to ovarian dysfunction, increasing the risk of infertility and affecting the hormonal balance crucial for sexual function. Studies indicate that paclitaxel, like cisplatin, can lead to early ovarian failure and a decline in estrogen production, which in turn impacts vaginal lubrication and sexual satisfaction [31]. Furthermore, recent research suggests that chemotherapy can lead to changes in the central nervous system, reducing arousal and orgasmic response [32]. The combination of chemotherapy and radiotherapy has been shown to exacerbate these effects, resulting in a higher incidence of sexual dysfunction compared to patients receiving only radiotherapy treatments [33]. These long-term effects significantly compromise the sexual health and quality of life of cancer survivors.

The main oncologic treatments for rectal and anal cancer, their pathophysiological mechanisms, associated sexual dysfunctions, prevalence data, and recommended management strategies are summarized in Table 1.

## 5. Psychological and Relational Aspects

Female sexual dysfunction following colorectal and anal cancer treatment is a multifaceted challenge, with psychological and relational factors playing a pivotal role in survivors’ quality of life. While the physical sequelae of oncologic therapies, such as surgical nerve damage, hormonal changes, and radiation-induced tissue injury, are well documented, emerging evidence increasingly highlights that psychological distress is a critical and often under-addressed contributor to sexual dysfunction in this population [34,35].

Recent studies in colorectal and anal cancer survivors report high prevalence rates of anxiety, depression, and post-traumatic stress symptoms, which are strongly correlated with sexual dysfunction and reduced sexual satisfaction. For example, a 2024 prospective cohort of female colorectal cancer survivors found that many experienced clinically significant depressive symptoms one year post-treatment (PHQ-9 ≥ 10, corresponding to at least moderate depression), with depression independently predicting poor sexual desire and arousal even after adjusting for physical impairments [36]. Benedict et al. Body image disturbance is a frequent consequence of cancer treatment, often linked to low self-esteem, shame, and avoidance of intimacy. This psychological distress negatively affects sexual well-being and quality of life, amplifying anxiety and depression while reducing desire and satisfaction [37].

Fecal incontinence, a common downstream consequence of sphincter-preserving surgery and pelvic radiation, is another significant psychological burden. In a multicenter cohort of 85 women, those with incontinence reported markedly higher sexual distress and social withdrawal compared to continent survivors, highlighting the interplay between physiological symptoms and mental health [38].

Pain is highly prevalent among cancer patients and closely associated with symptoms of PTSD, depression, and psychological distress, highlighting a bidirectional interplay that worsens quality of life [39].

Relational dynamics further compound these challenges. Cancer and its treatments can profoundly affect couples’ relationships, causing anxiety and intimacy challenges. Involving partners in sexual health discussions and providing couple-centered care can strengthen emotional support and improve quality of life. Healthcare professionals should be trained to address these issues to facilitate holistic cancer care [40]. Couple-based interventions in cancer care provide modest benefits for patients’ physical health and partners’ sexual relationship quality, while effects on sexual function and mental health remain uncertain. Integrating psychoeducation, skills training, and counselling supports communication and joint coping [41]. A study from 2020 showed that couples’ emotional intimacy substantially decreased post-treatment, mediated by mutual avoidance of sexual discussions and fear of upsetting partners [42]. Chronic stress from cancer surveillance and fear of recurrence also contribute to persistent hypoactive sexual desire and reduced sexual activity [43].

Encouraging intervention data exists: cognitive-behavioral therapy, mindfulness, and couple-based counseling have shown significant improvements in sexual function and relational satisfaction when tailored to colorectal and anal cancer survivors [44]. Yet, despite growing evidence, psychological assessment and intervention remain inconsistently implemented in clinical practice [45].

The NCCN Survivorship Guidelines (2024) explicitly recommend multidisciplinary approaches that integrate psychological and relational care alongside physical symptom management, emphasizing early mood-disorder screening and sexual health counseling [17]. The ESMO Guidelines recommend systematic screening with validated tools throughout the cancer trajectory to enable early detection and intervention. An integrated approach combining psychotherapeutic strategies, such as cognitive behavioural and mindfulness-based therapies, with pharmacological treatments including SSRIs and SNRIs, is advised to enhance symptom management and patient outcomes [46]. Regarding epidemiological data, in population-based Danish data (*n* = 2402), women with a permanent stoma exhibited almost threefold increased risk of overall sexual dysfunction (OR 2.95, 95% CI 1.05–6.38) and significantly higher dyspareunia and vaginal narrowing, even after controlling for radiotherapy [47]. In broader European cohorts, colorectal cancer survivors report substantially higher rates of vaginal dryness (28–35% vs. 5%) and dyspareunia (9–30% vs. 0%) compared to normative controls [48]. In many cultural contexts, sexual taboos and stigma continue to limit open dialogue on sexual health, a problem often exacerbated by the absence of integrated survivorship care. Socioeconomic status and religious affiliation likely act as significant covariates, shaping both the persistence of sexual tabooing and the pathways toward its removal [47]. Women often face shame and silence around sexual issues, which negatively impact their quality of life.

Italian qualitative studies [46] highlight similar challenges, including stigma related to the stoma and communication barriers with healthcare providers, resulting in social withdrawal and suspension of sexual activity in approximately 40% of female survivors. Other data [44] further emphasize the high burden of sexual dysfunction, especially in younger women, where more than 80% report dysfunction, and psychological distress is strongly linked to diminished quality of life and reduced social support. Unfortunately, sexual health remains under-discussed in clinical settings, perpetuating relational strain and emotional distress.

Collectively, these data highlight that psychological suffering related to body image alteration, stoma-related stigma, incontinence, and inadequate support profoundly impacts sexual desire, activity, and satisfaction, often independent of physical symptoms. Cultural norms surrounding modesty and shame further inhibit help-seeking, reinforcing relational strain and intimacy disruption. Recognizing and addressing this psychological and social dimension is therefore critical in improving sexual and overall quality of life among survivors.

Future research should prioritize the inclusion of underrepresented populations, including same-sex couples, and employ flexible delivery formats to enhance accessibility and participation. Moreover, the systematic involvement of partners within cancer care pathways may substantially improve quality of life and relational outcomes, supporting a more holistic and patient-centered model of oncology care [49].

In low- and middle-income countries, where survivorship services are scarce, non-profit organizations could help address critical gaps by fostering culturally sensitive education and opening dialogue on sexual health. However, without structural integration into healthcare systems, their impact remains limited, and stigma and inadequate support continue to undermine women’s sexual well-being and overall quality of life.

## 6. Assessment Tools and Limitations in Oncologic Female Populations

Accurate evaluation of sexual function in women with cancer is essential for understanding treatment impacts and guiding supportive interventions. Several patient-reported outcome measures have been developed to assess sexual health, quality of life, and related domains in female populations.

The Female Sexual Function Index (FSFI) is a 19-item self-report questionnaire designed to assess female sexual function across six domains: desire, arousal, lubrication, orgasm, satisfaction, and pain. Initially developed for the general population, the FSFI has demonstrated strong psychometric properties among female cancer survivors, correlating with measures of depression, menopausal symptoms, and quality of life. It serves as a valuable tool for monitoring sexual function and identifying cancer-related sexual dysfunction, facilitating early and targeted interventions within oncologic care pathways [50]. The Patient-Reported Outcomes Measurement Information System (PROMIS) is a validated NIH system to measure patient-reported outcomes (physical function, fatigue, pain, anxiety, depression) in cancer patients using short forms and computer-adaptive tests. It ensures precise, low-burden assessments, supports symptom monitoring, and enables comparisons with reference populations for patient-centered oncology care [51].

Corrigan et al. conducted a meta-analysis using validated PROs showing that patients with SCCA treated with CRT experience significant long-term sexual dysfunction. Although over half remained sexually active, women had a median FSFI score of 20.2, indicating moderate dysfunction, and men had a median IIEF-5 score of 14, reflecting mild to moderate erectile dysfunction. Younger patients were more likely to remain sexually active. These results highlight the persistent sexual health burden after CRT and the importance of developing interventions to mitigate these toxicities in long-term survivors [52].

Despite the availability of patient-reported outcome (PRO) tools to assess sexual function, their use in post-treatment evaluations for rectal and anal cancer survivors remains challenging. These challenges include variability in administration, interpretation, and integration into survivorship care. Additionally, there is a lack of PRO tools specifically validated for women treated for pelvic cancers, limiting accurate assessment and tailored interventions for sexual dysfunction in this population. Addressing these gaps is crucial to advancing patient-centered care and improving long-term quality of life outcomes for survivors.

## 7. Therapeutic and Rehabilitative Strategies

Combined effects of surgical trauma, pelvic radiotherapy-induced fibrosis, and systemic oncologic treatments, including chemotherapy and immunotherapy, drive FSD following rectal and anal cancer therapy. Traditional chemotherapies like oxaliplatin and 5-fluorouracil are associated with neurotoxicity and pelvic nerve injury, leading to diminished genital sensation, dyspareunia, and reduced arousal and orgasmic response [53,54]. Systemic side effects such as fatigue, hormonal imbalance, and mucosal inflammation further impair sexual health [55].

Recent oncology breakthroughs have introduced immunotherapies with potentially lower sexual toxicity. This represents a paradigm shift for patients who, before the advent of immunotherapy, were primarily managed with standard modalities such as chemoradiotherapy and surgery, both of which are well-documented to cause significant sexual side effects. In contrast, current evidence suggests that such adverse outcomes are not typically observed with immunotherapeutic approaches. In mismatch repair–deficient (dMMR) locally advanced rectal cancer, dostarlimab (PD-1 inhibitor) has demonstrated striking efficacy, with 100% of evaluable patients achieving clinical complete response after 6 months of treatment without needing chemotherapy, radiotherapy, or surgery [56], thus sparing patients the morphological and neurovascular damage linked to these modalities. The treatment was associated with only mild to moderate adverse events (rash, pruritus, fatigue, nausea), with no grade 3 or higher toxicities reported [57]. These outcomes suggest a markedly reduced risk of sexual dysfunction due to preservation of pelvic anatomy and avoidance of cytotoxic exposure.

In squamous cell carcinoma of the anal canal (SCAC), retifanlimab—another PD-1 inhibitor—evaluated in platinum-refractory disease (POD1UM-202) showed an objective response rate of ~13.8%, disease control rate of ~48.9%, and median duration of response ~9.5 months, with an acceptable safety profile and only ~11.7% grade 3 or higher treatment-related adverse events [58]. Subsequent phase 3 POD1UM-303/InterAACT-2 trial combining retifanlimab with carboplatin-paclitaxel in advanced SCAC improved progression-free survival (~9.3 vs. 7.4 months) and overall survival (~29.2 vs. 23.0 months) compared to chemotherapy alone [59], while maintaining a manageable toxicity profile. Although direct data on sexual function are still lacking, the absence of cumulative neuropathy and reduced mucosal damage compared to cytotoxic regimens suggests a lower risk for FSD.

Supportive pharmacologic interventions, such as topical estrogens and lubricants, remain critical for symptom control, especially in patients with mucosal changes [60,61]. Emerging non-hormonal options, including vaginal moisturizers and selective estrogen receptor modulators (SERMs), are essential for immunotherapy-treated patients who may have contraindications to hormone therapy [62].

Non-pharmacological rehabilitation, vaginal dilators, and pelvic floor physical therapy continue to play a vital role in preventing vaginal stenosis and enhancing pelvic muscle function post-treatment [62,63]. These strategies yield optimal results when tailored to the individual surgical and radiotherapy context.

We propose the implementation of psychosexual counseling and multidisciplinary survivorship programs to better address the emotional, body-image, and relational dimensions of female sexual dysfunction. Such structured interventions may enhance adherence to rehabilitation pathways and ultimately improve overall quality of life among survivors. Proactive patient education about the different sexual side effect profiles of chemotherapy versus immunotherapy supports early referrals and personalized management.

## 8. Discussion

This review highlights that female sexual dysfunction is not a marginal issue but a frequent and multifactorial outcome of rectal and anal cancer treatment. Hormonal withdrawal, autonomic nerve injury, and radiation-induced fibrosis profoundly disrupt sexual physiology.

At the same time, psychological distress, altered body image, and relational strain amplify the burden, especially among women with stomas or incontinence. Standard treatments such as TME, APR, LAR, radiotherapy, and chemoradiotherapy remain central to oncologic management, yet they carry predictable and under-addressed consequences for sexual health. Interventions are largely supportive and inconsistently implemented, while assessment tools like FSFI and PROMIS, although widely used, remain inadequate for pelvic cancer survivors.

Simultaneously, emerging immunotherapies offer the possibility of reducing long-term toxicity, but the absence of dedicated studies on sexual outcomes represents a missed opportunity.

Geographic and cultural disparities further expose how stigma and limited survivorship services perpetuate unmet needs. It is therefore essential that sexual health endpoints be systematically integrated into pelvic oncology trials and explicitly mandated in survivorship guidelines, ensuring that this fundamental dimension of quality of life receives the same level of rigor as oncologic outcomes.

## 9. Strengths and Limitations

This review offers a comprehensive overview of the current literature on sexual dysfunction in female survivors of rectal and anal cancer, highlighting an often under-recognized but clinically relevant issue. One of the main strengths lies in the focused approach on a specific patient population, which allows for an in-depth analysis of risk factors, treatment-related effects, and patient-reported outcomes. Furthermore, the inclusion of both physical and psychosocial dimensions of sexual health provides a more holistic perspective. However, some limitations should be acknowledged. The available literature is limited in scope and often based on small sample sizes or retrospective data, which may affect the generalizability of the findings. Additionally, variability in outcome measures and a lack of standardized assessment tools across studies make it difficult to draw firm conclusions. Future research should aim to address these gaps through prospective, longitudinal studies and the development of validated tools specifically tailored to this population. Current instruments, such as the FSFI and PROMIS SexFS, provide only a partial view of sexual health in colorectal and anal cancer survivors. The FSFI overlooks treatment-specific effects (e.g., stoma stigma, vaginal stenosis, incontinence) and excludes women who are sexually inactive due to therapy. At the same time, PROMIS lacks sensitivity to cancer-related impairments and relational or cultural influences. These shortcomings underline the need for an oncology-specific tool that integrates physical, psychological, and relational domains, includes sexually inactive women, and is adaptable across cultures. Such an instrument would improve assessment accuracy and guide survivorship interventions.

## 10. Conclusions and Future Perspective

FSD is a frequent and multifaceted consequence of treatment for rectal and anal cancers, arising from the combined effects of surgical trauma, pelvic radiotherapy, systemic therapies, and psychological distress. The literature consistently highlights the widespread occurrence of symptoms such as decreased libido, vaginal dryness, dyspareunia, and reduced sexual satisfaction, with significant repercussions on quality of life. Despite this, sexual health remains insufficiently addressed in routine clinical practice, with limited use of validated female-specific assessment tools and a lack of standardized management pathways. While instruments such as the FSFI are widely employed, they do not fully capture the complexity of dysfunction in women treated for pelvic malignancies. Moreover, most available evidence derives from retrospective or cross-sectional studies, with a notable scarcity of longitudinal research specifically focused on female populations. Psychological and relational dimensions, including those linked to stoma, body image, and partner dynamics, further amplify the burden and remain under-evaluated. The emergence of immunotherapy-based approaches, particularly in mismatch repair-deficient rectal cancer and advanced squamous cell anal carcinoma, opens new perspectives for organ preservation and potentially reduced sexual toxicity. However, dedicated studies exploring sexual health outcomes are still lacking. Future priorities should include the development of standardized gender-sensitive assessment tools, the systematic incorporation of sexual health endpoints into prospective observational and interventional trials, and the integration of sexual health into survivorship guidelines. Greater awareness, culturally adapted care models, and multidisciplinary collaboration will be crucial to improve the quality of life and address the unmet needs of female survivors of rectal and anal cancer.

## Figures and Tables

**Figure 1 cancers-17-03150-f001:**
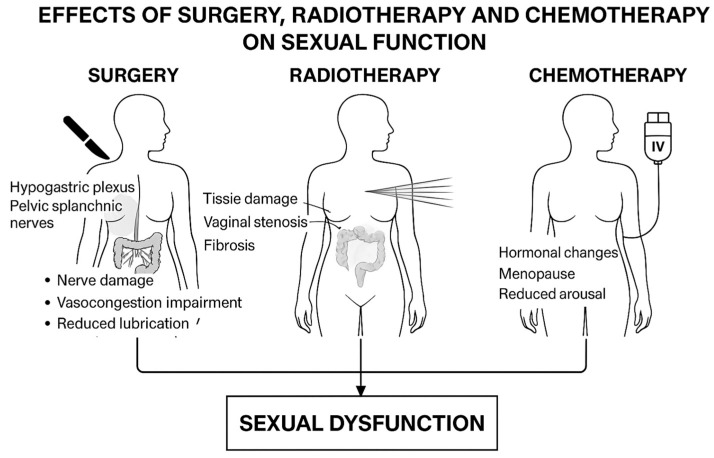
Effects of surgery, radiotherapy, and chemotherapy on sexual function.

**Table 1 cancers-17-03150-t001:** Oncologic treatments for rectal and anal cancer in women: mechanisms, prevalence, and management of sexual dysfunction.

Treatment	Pathophysiological Mechanisms	Main Reported Dysfunctions	Prevalence Data	Management and Rehabilitation Strategies
**Surgery (LAR, APR, TME)**	Injury to hypogastric plexus and pelvic autonomic nerves; anatomical alterations; stoma presence	Reduced lubrication, dyspareunia, decreased desire, loss of genital sensitivity, altered body image	19–62% postoperative dysfunction; up to 40% cessation of sexual activity	Nerve-sparing techniques, structured preoperative counseling, pelvic floor physiotherapy, psychosexual support
**Radiotherapy/Chemoradiotherapy**	Tissue fibrosis, hypoxia, reduced vascularization, mucosal atrophy; late neuropathy	Vaginal dryness, stenosis, dyspareunia, reduced elasticity, pain	Vaginal dryness: 50% vs. 24% (RT+surgery vs. surgery alone); vaginal stenosis up to 88%	Vaginal dilators, regular sexual activity, moisturizers/lubricants, local estrogen therapy (if not contraindicated)
**Chemotherapy**	Premature menopause due to ovarian toxicity; peripheral neuropathy (oxaliplatin, taxanes); endocrine alterations	Reduced desire, impaired lubrication, infertility, dyspareunia, diminished orgasmic response	>60% of survivors with FSFI < 26.5; persistent dysfunction common	Hormonal replacement (if indicated), endocrine support, cognitive-behavioral therapy (CBT), mindfulness, sexual rehabilitation
**Immunotherapy (dMMR rectal cancer, advanced SCAC)**	Organ-preserving approach; lower cumulative toxicity; preservation of pelvic anatomy	Direct data lacking; expected lower risk of dysfunction	Early trials (dostarlimab, retifanlimab): high response rates, no ≥G3 toxicities, no sexual function data	Prospective monitoring needed; patient counseling and sexual health follow-up recommended

Abbreviations: LAR = low anterior resection; APR = abdominoperineal resection; TME = total mesorectal excision; CRT = chemoradiotherapy; dMMR = mismatch repair deficient; SCAC = squamous cell carcinoma of the anal canal; FSFI = Female Sexual Function Index.

## Data Availability

No new data were created or analyzed in this study. Data sharing is not applicable to this article.

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
