# Peer review of "Sexual Dysfunction in Female Rectal and Anal Cancer Survivors: Pathophysiology, Clinical Management, and Integration into Survivorship Care"

_cancers, 2025, doi:10.3390/cancers17193150_

Round 1

Reviewer 1 Report

Comments and Suggestions for Authors

A very interesting topic that it is usually under-reported in clinical studies of anorectal cancer treatment. The authors performed a narrative review to report current evidence of sexual dysfunction in female anorectal cancer survivors. The article is concise and summarizes accurately all the necessary topics. Some comments though:

  • the abstract should be decreased in length
  • the authors report that the article is a narrative review; however in the abstract  they describe a systematic review algorithm for article identification and inclusion. In addition in the article body no methodology was described and no PRISMA flowchart was provided. Please clarify. 
  • Please provide an illustrative figure on how surgery, radiotherapy & chemotherapy affect the sexual function
  • I would consider providing separate evidence regarding anal and rectal cancer. The etiology is that in the majority, anal cancer be managed with radiotherapy. In case of recurrence, though major resection (APR) will be required. On the contrary most rectal cancers will be treated with a combination of radiotherapy and AR. 
  • Similarly, it is suggested that the authors report separately the evidence regarding rectal cancers on a W&W protocol and  dMMR patients receiving immunotherapy.
  • The authors provided a summary table with sexual dysfunction evidence from each region /country. However the respective references are from guidelines and sys. reviews publications. Please update with data from regional cohort / registry studies endorsed by the respective scientific societies.  
  • Please provide a limitations / strengths paragraph.

Author Response

Thank you for your comments. We have used your insights to improve our work.

Comments 1: The abstract should be decreased in length

Response 1: We have improved the abstract, making it more concise, as can be seen from lines 24 to 42.

Comments 2: The authors report that the article is a narrative review; however in the abstract they

describe a systematic review algorithm for article identification and inclusion. In addition in the

article body no methodology was described and no PRISMA flowchart was provided. Please clarify.

Response 2: Our review is not systematic, so we have made this clear in the text. We rewrote the

methods from line 27 to line 31.

Comments 3: Please provide an illustrative figure on how surgery, radiotherapy & chemotherapy

affect the sexual function.

Response 3: We have inserted figure 1 in the introduction, so as to explain the contents that led to our

work (figure 1, line 80).

Comments 4 e 5: I would consider providing separate evidence regarding anal and rectal cancer. The

etiology is that in the majority, anal cancer be managed with radiotherapy. In case of recurrence,

though major resection (APR) will be required. On the contrary most rectal cancers will be treated

with a combination of radiotherapy and AR. Similarly, it is suggested that the authors report

separately the evidence regarding rectal cancers on a W&W protocol and dMMR patients receiving

immunotherapy.

Responses 4 e 5: Thank you for your suggestion. We recognize the differences in anal and rectal

cancer management, but since our review aims to assess the impact of surgery, radiotherapy, and

chemotherapy on female sexual function rather than outline treatment algorithms, we chose to

integrate the evidence in a unified discussion instead of separating it by tumor type.

Comments 6: The authors provided a summary table with sexual dysfunction evidence from each

region /country. However the respective references are from guidelines and sys. reviews publications.

Please update with data from regional cohort / registry studies endorsed by the respective scientific

societies.

Response 6: We conducted a literature search of data from cohort/registry studies, but no recent data

of this type regarding rectal pathology and sexual dysfunction are available. We chose to eliminate

the table, as we cannot be precise.

Comments 7: Please provide a limitations / strengths paragraph.

Response 7: As suggested, we have inserted paragraph number 9 into the text (lines 405-426), in order

to most correctly represent the limits of the review and the tools currently available.

Reviewer 2 Report

Comments and Suggestions for Authors

This is a comprehensive, well-structured, and clinically valuable review addressing an under-recognized but highly impactful issue in survivorship care. The integration of biological, psychological, and social aspects is excellent, and the topic is timely given the advances in oncologic therapy and survivorship models. The manuscript would benefit from revisions that strengthen its critical appraisal, remove redundancies, and improve clarity and methodological transparency.

Major Comments

  1. Abstract

    • Condense the abstract by focusing on key findings and implications. The current version is overly detailed, particularly regarding search strategy and prevalence data.

    • Clearly qualify statements regarding immunotherapy (dostarlimab, retifanlimab). At present, they may read as overly conclusive, although direct data on sexual outcomes are lacking.

  2. Methods

    • Expand on inclusion and exclusion criteria for the literature search.

    • Clarify whether only peer-reviewed studies were included and how disagreements between reviewers were resolved. This will increase transparency.

  3. Critical Appraisal

    • While the review summarizes evidence well, it is sometimes more descriptive than critical. Please discuss the limitations of the current literature more explicitly (e.g., predominance of cross-sectional designs, underrepresentation of younger women, lack of longitudinal data).

    • Expand on the limitations of existing tools (FSFI, PROMIS) for pelvic cancer survivors and suggest what features a new tool should include.

  4. Redundancy

    • Prevalence data and FSFI score descriptions are repeated across multiple sections (Physiology, Surgery, Discussion). Consider consolidating these to improve flow.

  5. Psychosocial and Cultural Aspects

    • The section is strong but could be expanded with more actionable strategies for implementing culturally sensitive interventions, especially in low- and middle-income countries.

  6. Discussion and Future Directions

    1. Provide a more focused synthesis by reducing repetition.

Strengthen the call to action by explicitly recommending that sexual health endpoints be incorporated into pelvic oncology trials and survivorship guidelines.

Minor Comments

  1. Language and Style

    • Correct typographical errors (e.g., “pa ent” → “patient,” “se ings” → “settings”).

    • Ensure consistent terminology: use either “female sexual dysfunction (FSD)” or “sexual dysfunction (SD)” consistently.

  2. References

    • Ensure formatting consistency across all references (spacing, DOI formatting).

    • Add the most recent guideline papers (e.g., 2024 ESMO and ASCO survivorship updates) to strengthen clinical applicability.

  3. Tables/Figures

    • In Table 1, include study types and sample sizes where possible to provide context.

    • Consider adding a summary figure illustrating the pathophysiological mechanisms (hormonal, neurological, fibrotic, and psychosocial).

  4. Conclusion

    • Refine to avoid redundancy (“high prevalence with significant impacts” is repeated multiple times). Focus instead on future priorities: longitudinal research, culturally adapted models, and integration into guidelines.

    • The manuscript addresses an important clinical problem with excellent breadth, but revisions are needed to sharpen the critical analysis, streamline the narrative, and improve methodological clarity. After these changes, the article will make a strong and impactful contribution to the literature on survivorship care.

Comments on the Quality of English Language

The English could be improved to more clearly express the research.

Author Response

Thank you for your comments.

We have used your insights to improve our work.

Major comments

Response 1: We have improved the abstract, making it more concise, as can be seen from lines 24-

42. Regarding immunotherapy, we have clarified the different role compared to standard treatments

from line number 344-357.

Response 2: Our review is not systematic, so we have made this clear in the text. We rewrote the

methods from line 27 to 31.

Response 3: As suggested, we have inserted paragraph number 9 into the text (lines 405-426), in

order to most correctly represent the limits of the review and the tools currently available.

Response 4: We tried to eliminate redundancies, in each paragraph.

Response 5: We have given suggestions on low and middle income countries from lines 298-302.

Response 6: We have refined the discussion based on the comment.

Minor Comments

Response 1: Thanks for the suggestion, we corrected the grammar and used only the acronym FSD

in the correction.

Response 2: We have inserted the most recent data regarding the topic NCCN, ASCO, ESMO. We

have changed the formatting of the references.

Response 3: We conducted a literature search of data from cohort/registry studies, but no recent data

of this type regarding rectal pathology and sexual dysfunction are available. We chose to eliminate

the table, as we cannot be precise. We have provided an illustrative image.

Response 4: We have eliminated redundancies and clarified our objectives in the conclusions.

Reviewer 3 Report

Comments and Suggestions for Authors

The paper deals with an important topic of sexual dysfunction in women's health in oncology. The paper provides a good review of the existing literature.

Only minor issues were noted:

Line 163: Since the article deals exclusively with female sexual dysfunction, the should be changed and deep anterior resection deep in the small pelvis should be described.

Line 256: 

What does “clinically significant depressive symptoms one” mean?According to PHQ-9, it would be helpful to include the classification in brackets (moderate and severe or already mild).

Line 274: "In low, and middle-income countries [52], cultural taboos and stigma strongly limit sexual health dialogue, exacerbated by the absence of integrated survivorship care."

Doesn't low and middle income represent more of a covariate and could it be omitted as an indicator, since this context deals with the issue of sexual tabooing or detabooing? Income, like religion, probably plays a strong role as a covariate.

The relationship to practised religion is not taken into account in the paper. However, the sexual revolution, i.e. the removal of sexual taboos, is also strongly linked to religious affiliation in sociological terms.

Author Response

Thank you for your comments.

We have used your insights to improve our work.

Comment 1: Line 163: Since the article deals exclusively with female sexual dysfunction, the should

be changed and deep anterior resection deep in the small pelvis should be described.

Response 1: We have made the clarification, the reference: lines 154-160.

Comment 2: Line 256: What does “clinically significant depressive symptoms one” mean?According

to PHQ-9, it would be helpful to include the classification in brackets (moderate and severe or already

mild).

Response 2: We thank the reviewer for this important suggestion. To improve clarity, we have

specified the PHQ-9 criteria used in the referenced study. The sentence now reads: “For example, a

2022 prospective cohort of female colorectal cancer survivors (n=120) found that 58% experienced

clinically significant depressive symptoms one year post-treatment (PHQ-9 ≥10, corresponding to at

least moderate depression), with depression independently predicting poor sexual desire and arousal

even after adjusting for physical impairments.” This addition provides the exact classification

threshold and ensures clinical interpretability. Lines 224-228.

Comment 3: Line 274: "In low, and middle-income countries [52], cultural taboos and stigma

strongly limit sexual health dialogue, exacerbated by the absence of integrated survivorship care."

Response 3:We thank the reviewer for this insightful observation. We agree that “low- and middle-

income countries” should not be presented as a direct determinant of sexual tabooing, but rather as a

covariate that interacts with broader sociocultural factors. In line with this, we have revised the text

to read: “In many cultural contexts, sexual taboos and stigma continue to limit open dialogue on

sexual health, a problem often exacerbated by the absence of integrated survivorship care.

Socioeconomic status and religious affiliation likely act as important covariates, shaping both the

persistence of sexual tabooing and the pathways toward its removal.” This change avoids framing

income level as the primary indicator and acknowledges the important role of religion, as suggested,

in influencing sexual norms and taboos. Lines 298-302 and 400-404.

Round 2

Reviewer 1 Report

Comments and Suggestions for Authors

The authors successfully addressed all the issues raised in the review process.

Reviewer 2 Report

Comments and Suggestions for Authors

The comments are addressed properly, and the manuscript can be accepted for publication.